# Local Deformation Behavior of the Copper Harmonic Structure near Grain Boundaries Investigated through Nanoindentation

**DOI:** 10.3390/ma14195663

**Published:** 2021-09-29

**Authors:** Viola Paul, Masato Wakeda, Kei Ameyama, Mie Ota-Kawabata, Takahito Ohmura

**Affiliations:** 1Department of Materials Science and Engineering, Graduate School of Engineering, Kyushu University, 774 Motooka, Nishi-ku, Fukuoka 819-0395, Japan; paul.viola@nims.go.jp; 2Research Center for Structural Materials, National Institute for Materials Science, 1-2-1 Sengen, Tsukuba 305-0047, Japan; wakeda.masato@nims.go.jp; 3Department of Mechanical Engineering, Faculty of Science and Engineering, Ritsumeikan University, 1-1-1 Noji-higashi, Kusatsu 525-8577, Japan; ameyama@se.ritsumei.ac.jp (K.A.); mie-ota@fc.ritsumei.ac.jp (M.O.-K.)

**Keywords:** pure copper, harmonic structure, nanoindentation, plastic deformation

## Abstract

The copper harmonic structure, which consists of a coarse-grained “core” surrounded by a three-dimensional continuously connected fine-grained “shell,” exhibits both high ductility and high strength. In the present study, dislocation interactions at the shell–core boundary in the copper harmonic structure were directly measured using nanoindentation and microstructural observations via kernel average misorientation (KAM) to further understand the reason for its excellent mechanical properties. KAM analysis showed that the dislocation density in the vicinity of the shell–core boundary within the core region gradually increases with increasing plastic strain. The variation in the nanohardness exactly corresponds to the KAM, indicating that the higher strength is primarily caused by the higher dislocation density. The critical load for nanoindentation-induced plasticity initiation was lower at the shell–core boundary than at the core–core boundary, indicating a higher potency of dislocation emission at the shell–core boundary. Because dislocation–dislocation interactions are one of the major causes of the increase in the flow stress leading to higher strain hardening rates during deformation, the excellent balance between strength and ductility is attributed to the higher potency of dislocation emission at the shell–core boundary.

## 1. Introduction

The fabrication of a bimodal microstructure called the “harmonic structure,” which consists of a coarse-grained “core” surrounded by a three-dimensional continuously connected fine-grained “shell,” was proposed by Ameyama et al. It has been reported that this microstructure possesses both higher strength and ductility compared with homogenous coarse-grained and fine-grained counterparts in various materials [1,2,3,4,5].

In general, the harmonic structure improves the mechanical properties of various materials. However, according to Orlov et al., specific properties (such as the total elongation) still depend on the type of material [6]. The total elongation decreases in pure iron, aluminum, and SUS304L stainless steel, while it remains unchanged in pure titanium and nickel. However, pure copper and SUS316L stainless steel exhibit an increase in total elongation [3,7]. For various materials with different elongations, a characteristic behavior of the stress–strain curve is the strain-hardening rate. For example, pure copper shows a higher strain-hardening rate up to fracture compared with other materials [3]. It is reasonable to assume that a higher strain rate leads to higher elongation or ductility based on the plastic instability condition [8,9]. However, the exact reasons for the higher strain rate and higher elongation have yet to be reported. The copper harmonic structure exhibits a combination of excellent ductility with improved mechanical strength compared with other materials with the same structure. Consequently, the copper harmonic structure was analyzed in this study to further understand the strengthening mechanisms of the harmonic structure.

One of the major factors related to the strengthening mechanism is the role of grain boundaries. The effects of grain boundaries are generally divided into two factors. On the one hand, grain boundaries act as barriers to dislocation glide to impede slip transfer at grain boundaries, which is widely accepted as a significant factor for the strengthening of polycrystalline metals. It has been reported that the fine-grained/coarse-grained boundary causes back-stress hardening, which contributes to the synergistic strength–ductility properties of the harmonic structure [10]. On the other hand, grain boundaries act as a source of dislocation generation, which assists plasticity. A dislocation event can be viewed as the activation of a dislocation source that lies within or near the boundary. The concept of grain boundaries acting as dislocation sources was first proposed by Li [11]. Since then, numerous studies have reported evidence of grain–boundary dislocation emission [12,13,14,15,16]. In fact, direct experimental observation by transmission electron microscopy was reported separately by Essmann et al. for copper [12], Li et al. for fine-grained interstitial-free (IF) steel [17], and Murr and Hecker for 304 stainless steels [18]. Ohmura et al. studied the mechanical behavior in the vicinity of a single grain boundary in an IF steel using a nanoindentation technique and found that the grain boundary acts as an effective dislocation source compared with the grain interior [19,20]. Hence, by incorporating insights from previous works, it is necessary to characterize the mechanical properties of the copper harmonic structure in a localized region, including grain boundaries, via small-scale testing methods to obtain more explicit evidence of the factors affecting strengthening. Particular attention should be given to the shell–core boundary because it is the most distinct microstructure in the harmonic structure.

Nanoindentation is a versatile technique for quantitatively analyzing the mechanical properties of materials at the nanoscale [21,22,23,24,25,26,27]. Therefore, to elucidate the mechanism responsible for the high elongation and corresponding high strength of the copper harmonic structure, the nanoindentation technique was used to quantitatively investigate the deformation behavior as well as microstructural characterization related to the shell–core boundary in this study.

## 2. Materials and Methods

Commercial pure copper was used as the starting material for the fabrication of the copper harmonic structures. The powder was mechanically milled and subsequently consolidated by spark plasma sintering. The detailed process conditions are reported elsewhere [3]. For nanoindentation and electron backscatter diffraction (EBSD) observations, the surfaces of the investigated samples were mechanically polished and electropolished in a 2:1:1 solution of phosphorous acid, ethanol, and distilled water at room temperature under an electric potential of 10 to 15 V to remove the damaged layer.

Nanoindentation experiments were performed using a Hysitron Triboindenter 950 (Bruker Co., Minneapolis, MN, USA) with a Berkovich indenter at a maximum peak load of 500 µN. The device was equipped with a scanning probe microscope (SPM) to obtain images of the sample surface before and after indentation testing. The investigated sample microstructure was analyzed using EBSD data obtained by field-emission scanning electron microscopy (SEM, Carl-Zeiss LEO-1550 Schottky, 20 kV operating voltage, Tokyo, Japan). The grain size and kernel average misorientation (KAM) maps were obtained from the EBSD data using the TSL orientation imaging microscopy (OIM) software (TSL Solutions Co., Sagamihara, Japan, version 7). Due to the finer grain size of the shell region compared with the core region, two step sizes were used to measure the grain size distribution by IQ image and KAM distribution; a lower resolution of 0.5 µm step size was used to determine the core region and a higher resolution of 0.02 µm step size was used to measure the finer grain size in the shell region. The KAM quantified in this study is the average misorientation around a measurement point with respect to a defined set of first neighboring six data points. The measured range of the KAM maps was 0° to 5°, which was expressed using a color scheme.

Two types of samples were used in this study, before and after the tensile test, and three positions were chosen on the post-tensile test sample with various local strains. Nanoindentation experiments were conducted on the core–core boundary, shell–core boundary, and core interior (which were determined by the KAM map) to evaluate the deformation behavior associated with the various regions of the microstructure. The critical load, $P_{c}$, for plasticity initiation was obtained from the loading curve, and the nanohardness, $H_{n}$, was calculated from the unloading curve.

## 3. Results and Discussion

The microstructural information of the copper harmonic structure before deformation is shown in Figure 1, which was obtained through SEM–EBSD analysis with 0.5 and 0.02 µm step sizes. The image quality (IQ) map with a 0.5 µm step size in Figure 1a demonstrates that the coarse-grained “core” (light gray area) was surrounded by the fine-grained “shell” network (dark contrast), as indicated on the image. To reveal the finer grain size in the shell region, the area in the yellow box in Figure 1a was scanned with the smaller step size (0.02 µm), and the IQ map is shown in Figure 1b. Figure 1c shows the distribution of the grain size diameter based on an equiaxial approximation obtained from the EBSD data in Figure 1a,b. The minimum value was set based on the step size in the EBSD scan, and the maximum value was set to the maximum grain size in the scanned area. Accordingly, a set of 20 geometrically spaced bins was created between 0.5 and 100 µm with a logarithmic scale on the x-axis. The average grain size of the core region was 18.0 µm (area fraction of 67%), which was measured by Gaussian fitting (red line in Figure 1c). Using a similar method, the grain-size distribution of the fine-grained region in Figure 1b was fit by a Gaussian distribution, which is shown by the green line on a linear scale in the inset of Figure 1c. The average grain size of the shell region was 0.4 µm (area fraction of 33%). Because the maximum grain size of the fine-grained region (Figure 1b) was 2.2 µm, that size was set as the border between the core and shell regions in the grain size distribution in (Figure 1c).

To compare the local strain distributions of the core and shell regions after the tensile test, three positions were investigated using SEM–EBSD. Figure 2a shows an SEM micrograph of the fractured specimen indicating the three positions: A, near the shoulder of the gauge portion; C, near the fracture surface; and B, intermediate between A and C. The IQ (top half) and KAM (bottom half) maps of the three positions as well as those of the pre-tensile test sample are shown in Figure 2b. Magnified KAM images around the shell–core boundary are also shown at the bottom of each image, showing the shell region blind to focus on the KAM within the core region near the shell–core boundary (individual grains are not shown for clarity owing to the fine and complicated structure in the shell). A confidence index lower than 0.1 is not shown in the KAM images [28]. Note that the shell region in the IQ image (dark contrast) could not be observed in Figure 2b with 0.5 µm step size of EBSD scan due to finer grain size. The strain distribution in the investigated positions was quantified via KAM, where the dislocation density is related to the strain [29]. Figure 2c shows the average KAM of the shell and core regions of the four positions in Figure 2b. The KAM distribution was plotted by highlighting several areas at shell and core regions in Figure 2b, then the average KAM of the areas for the respective regions were then acquired. Unlike the core region which can be observed with broad areas using lower spatial resolution of 0.5 µm step size, the KAM of the shell region was obtained by EBSD scan of 0.02 µm step size to detect the finer grain size in the shell region. There was a steady increase in the average KAM of the core region from A to C. In contrast, the KAM in the shell region was almost constant. Because a higher KAM can correspond to a higher dislocation density, this finding is consistent with previous studies, which showed that dislocations were generated more easily in the softer region, which is the core region in this case [30,31,32].

To reveal the spatial distribution of the KAM, particularly near the shell–core boundary within the core region, the local average KAM are plotted as a function of the distance from the shell–core boundary for each of the four positions in Figure 2d. The local average KAM was evaluated within a 40 × 40 µm^2^ square every 5 µm from the shell–core boundary toward the core grain center for a total distance of 30 µm. The KAM was constant for the pre-tensile test sample, while for the post-tensile test sample, the KAM gradually decreased with increasing distance from the shell–core boundary. This indicates a higher strain near the shell–core boundary. As mentioned previously, because a higher strain is associated with a higher dislocation density, this result suggests the dislocation density was higher near the shell–core boundary. Furthermore, the dislocation density near the shell–core boundary increased from A to C, even at the farthest distance, which is consistent with the trend in Figure 2c.

Figure 3a shows the IQ maps overlaid on the grain size maps near the shell–core boundary before and after the tensile test. The color scheme indicates the shell (green) and core (red), and the array of gray dots are the nanoindentation marks. The KAM maps are shown at the bottom of each image. Figure 3b shows the $H_{n}$ of the four regions as a function of the distance from the shell–core boundary. $H_{n}\;$ slightly decreased with increasing distance, except before the tensile test, which is the same trend as that shown in Figure 2d. This hardness distribution can likely be attributed to the interaction between the indentation-induced dislocations underneath the indenter and pre-existing dislocations near the shell–core boundary, which led to dislocation hardening [21]. In areas with fewer dislocations, the indentation-induced dislocations could easily move without any dislocation obstacles. Hence, $H_{n}$ was the lowest before the tensile test. Additionally, position A exhibited the steepest rate of decrease among the three positions after the tensile test, as shown in Figure 3b. To consider the difference in the rate of decrease, the normalized KAM is plotted as a function of the distance from the shell–core boundary in Figure 3c. The normalized KAM was calculated using Figure 2d, where the KAM at every distance was divided by the value at the farthest distance. The rate of decrease was the steepest in position A, suggesting the highest rate of decrease in dislocation density with the distance from the shell–core boundary. Therefore, it can be assumed that the larger rate of decrease in $H_{n}$ in position A was due to a larger change in dislocation density with distance.

To further understand the reason for the higher dislocation density near the shell–core boundary, we investigated the potency of dislocation generation at the shell–core boundary. Nanoindentation was performed on the core–core and shell–core boundaries to quantitatively analyze the plasticity initiation behavior. Figure 4a shows the KAM (top) and grain size (bottom) maps overlaid on the IQ map in the nanoindentation area, which consists of shell and core regions. The position of the selected area on the post-tensile test sample is shown by the yellow square in the SEM image in the top-left inset of Figure 4a. The IQ image in the yellow box in Figure 4a is magnified in Figure 4b to show the core–core and shell–core boundaries more clearly, and Figure 4c shows the SPM image of the same region. The dotted white line and solid black line indicate the shell–core and core–core boundaries, respectively. Nanoindentation was performed almost exactly on the core–core and shell–core boundaries, as indicated by the yellow arrows in Figure 4c. The typical load–depth (*P*–*h*) curves are shown in Figure 4d. The initial loading curves fit well with the dotted line, which was calculated using Hertz contact theory [33]:(1)$$P=\frac{4}{3}E_{r}R^{1/2}h^{3/2}$$
where *P* is the applied load, $E_{r}\;$ is the reduced modulus, R is the indenter tip radius, and *h* is the penetration depth. Pop-in phenomena can be observed as sudden displacement bursts, which appear as a plateau in the loading curve. The pop-in phenomenon can be understood as dislocation nucleation and propagation under an initial elastic strain and requires shear stresses in the range of the theoretical strength of a defect-free metal [34,35,36,37,38]. Therefore, the maximum shear stress, $\tau_{max}$, underneath the indenter during loading is given as [33]
(2)$$\tau_{max}=0.18{\left(\frac{E_{r}}{R}\right)}^{2/3}P^{1/3}$$

The value of $\tau_{max}$ at the onset of plasticity can be obtained by substituting the value of $P_{c}\;$ for P in Equation (2), which reflects the critical shear stress for dislocation nucleation. The calculated critical shear stress before the tensile test in the core grain interior was 2.82 GPa. Because the shear modulus, *G*, of pure copper is approximately 48.3 GPa [39], the calculated shear stress is in the range of the theoretical strength (G/2π–G/30).

The *P*–*h* curve in Figure 4d exhibits small and large pop-in phenomena for the core–core and shell–core boundaries, respectively. In the present case, a clear large pop-in phenomenon is detected at 1 nm threshold of excursion depth. However, regardless of the scale of the pop-in event, the deviation of the loading curve from the Hertz fit indicates a transition from elastic to plastic behavior. Hereafter, the transition load at which the pop-in phenomenon occurs is defined as $P_{c}$, as indicated by the arrows in Figure 4d.

The average $P_{c}$ for plasticity initiation is shown in Figure 4e, indicating a lower value for the shell–core boundary than for the core–core boundary, although the difference is not very large. This reveals that plastic initiation occurs more easily in the shell–core boundary than in the core–core boundary. If an applied load causes a homogenous stress at both the core–core and shell–core boundaries, it can be presumed that the shell–core boundary will preferentially generate dislocations. The higher potency of dislocation generation at the shell–core boundary could be a major reason for the higher dislocation density in the vicinity of the shell–core boundary, as shown in Figure 3.

The higher potency of dislocation generation at the shell–core boundary is supported by the magnified KAM maps in Figure 5. Figure 5a–c show KAM maps near the shell–core boundary before the tensile test and at positions A and B after the tensile test, respectively. FG refers to the fine grains in the shell region, and CG refers to the coarse grains in the core region. The dashed white line and the thin black line indicate the shell–core and core–core boundaries, respectively. The black regions in each figure referred to the FG (grain size below 2.2 µm), which are not indexed to reveal the KAM map within the core region more clearly. Furthermore, the confidence index lower than 0.1 is also not shown in the KAM images. As shown in Figure 5b, position A (which had a lower strain) exhibited a higher KAM near the shell–core boundary than at the core–core boundary. As mentioned previously, a high KAM can be an indication of a high dislocation density. Because plasticity initiation is relatively easier at the shell–core boundary than at the core–core boundary (as described in Figure 4), the higher KAM observed in the area nearest to the shell–core boundary at position A could be evidence of dislocation emission from the shell–core boundary. This phenomenon was also reported by Shin et al. for heterogeneous β-Ti alloys, wherein the plastic strains were enhanced near the interfaces separating coarse-grained/fine-grained regions [40]. Consequently, this dislocation emission in the shell–core vicinity may lead to local plastic deformation. This may result in a significant strain-hardening effect, contributing to an increase in the total elongation and strength of the copper harmonic microstructure. Eventually (as shown for the higher strain at position B), dislocation emission was not only observed at the shell–core boundary but also at the core–core boundary, as indicated by the white arrows in Figure 5c. As shown in Figure 2d, the KAM increased from position A to position C, even in the deep interior of the core region. Hence, the higher dislocation density at positions B and C could be attributed to dislocation generation at not only the shell–core boundary but also at the core–core boundary.

Park et al. reported that the shell–core boundary plays an important role in the strength–ductility balance in SUS304L stainless steel owing to back stress-hardening [30]. They found that the pile up of exerted geometrically necessary dislocations at the interface between coarse-grained/fine-grained regions causes back-stress hardening, which is responsible for the synergistic strength and ductility balance of the harmonic structure. With the distinct dislocation emission results obtained in the present research, another possible factor is proposed in addition to the back-stress hardening reported by Park et al., which contributes to the strength–ductility balance of the harmonic structure. Dislocation emission at the shell–core boundary may increase dislocation–dislocation interactions. Accordingly, this can increase the strain-hardening rate, which could also lead to the strength–ductility synergy observed for the copper harmonic structure.

Figure 6a shows a schematic of the true stress–true strain curve for homogenous materials [9]. Generally, metals undergo strain hardening during deformation, which may assist in bearing loads during deformation. It is notable that necking or localized deformation, which is accompanied by a reduction in the cross-sectional area of the tensile specimen, occurs when the increase in stress is higher than the increase in the load-carrying ability of the metal due to strain hardening [8]. Therefore, as shown in Figure 6a, the point of necking under tensile stress is obtained at the point where the strain rate hardening equals the stress (dσ/dε  ≤ σ) in the true stress–true strain curve [9]. Typically, homogenous materials exhibit high strength but poor ductility owing to their poor strain-hardening capabilities, as indicated in Figure 6a. Therefore, a larger strain-hardening rate can delay necking and provide additional elongation to achieve simultaneous high strength and high ductility [8], as schematically shown in Figure 6b. Because the shell–core boundary had a higher potency of dislocation generation, the $H_{n}$ in the shell–core vicinity was higher closer to the shell–core boundary, as shown in Figure 3b. This is presumably due to the interaction between the indentation-induced dislocations and the tensile stress-induced dislocations near the shell–core boundary. Considering macroscopic deformation at a certain tensile strain, the density of pre-existing dislocations (previously induced by tensile stress) is higher near shell–core boundary than in the core interior. Additionally, when the strain increases slightly, more dislocations are generated at the shell–core boundary than in the core interior, leading to a higher strain-hardening rate near the shell–core boundary. Therefore, the high potency of dislocation generation at the shell–core boundary is a major factor for the combined high elongation and high strength of the copper harmonic structure. The probable reason for the high potency at the shell–core boundary might be due to the unstable high-energy grain boundary structure than core–core boundary [41,42]. Since the fine grains in the shell region were generated by mechanical milling, the grain boundary might be unstable, such as an ultra-fine grain produced by a severe plastic deformation [43,44]. Therefore, further work is required to investigate the shell-core boundary character in copper harmonic structure.

## 4. Conclusions

In this study, the local mechanical behavior associated with the strain distribution in the vicinity of the shell–core boundary in the copper harmonic structure was studied using nanoindentation and microstructural observations via KAM analysis based on SEM–EBSD. The KAM increased with the nominal plastic strain in the deformed tensile specimen, and a higher KAM was observed near the shell–core boundary in the core region. The $H_{n}$ in the shell–core vicinity was found to increase with increasing KAM for every position. Because a higher KAM corresponds to higher dislocation density, the higher $H_{n}$ is attributed to strain hardening caused by the interaction between the pre-existing tensile stress-induced dislocations and the indentation-induced dislocations. When indentations were made exactly on the grain boundaries, the average $P_{c}$ for plasticity initiation was lower for the shell–core boundary than for the core–core boundary in the core interior. Therefore, it can be presumed that the shell–core boundary preferentially generates dislocations during deformation, which is consistent with the higher KAM near the shell–core boundary. Because dislocation–dislocation interactions are a major factor contributing to the increase in the flow stress (leading to a higher strain hardening rate during deformation), the higher potency of dislocation emission at the shell–core boundary is one of the major factors contributing to the excellent balance between the strength and ductility of the copper harmonic structure.

## Figures and Tables

**Figure 1 materials-14-05663-f001:**
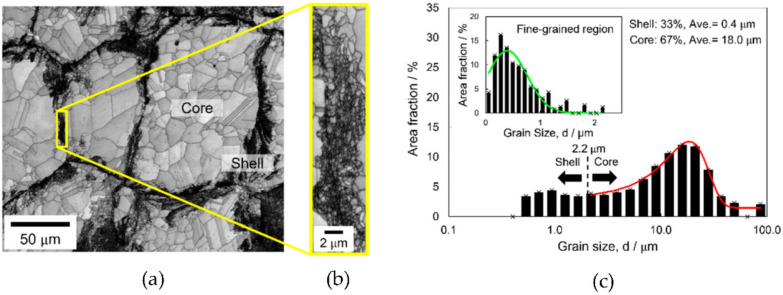
(**a**) Image quality (IQ) map of the copper harmonic structure obtained by scanning electron microscopy (SEM)–electron backscatter diffraction (EBSD) with a 0.5 µm step size. (**b**) Magnified image of the shell region indicated by the yellow box in (**a**) with a 0.02 µm step size. (**c**) Grain size distributions of the copper harmonic structure and fine-grained region (inset); the average grain size and area fraction of the shell and core regions are shown in the top right.

**Figure 2 materials-14-05663-f002:**
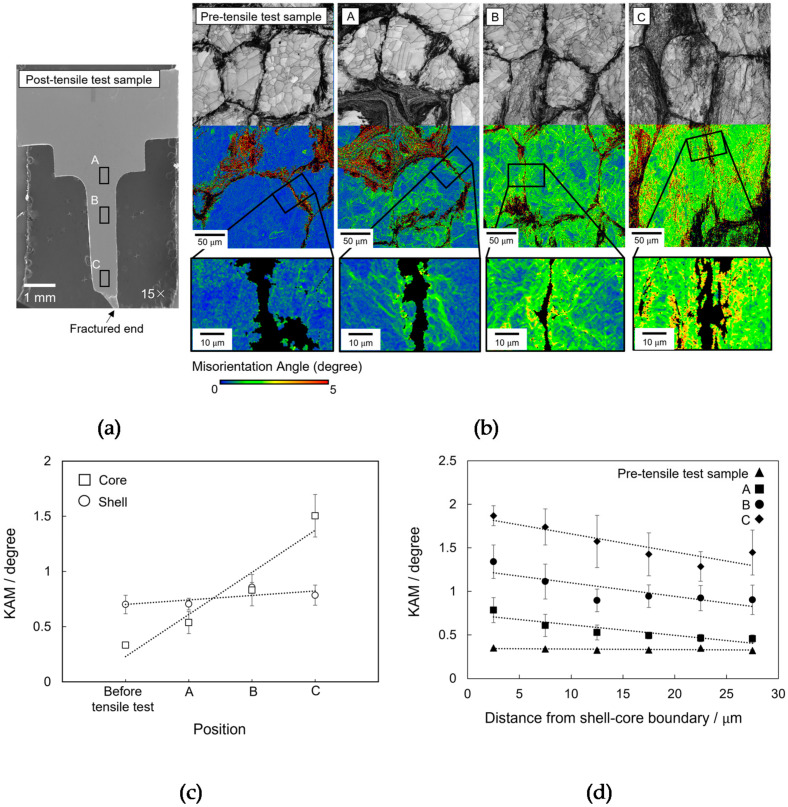
(**a**) SEM micrograph showing the different investigated positions of the post-tensile test sample (labeled A, B and C). (**b**) Variation in the IQ (top half) and kernel average misorientation (KAM) (bottom half) with a 0.5 µm step size of the pre-tensile test sample and of the post-tensile test sample at the different position shown in (**a**). Magnified KAM images with the same step size of the marked boxes near the shell–core boundary are also shown, in which the shell region is not indexed (individual grains are not shown for clarity owing to the fine and complicated structure). (**c**) Average KAM of the shell and core regions in the positions shown in (**b**). (**d**) Spatial distribution of the KAM for the four positions plotted as a function of the distance from the shell–core boundary toward the grain center.

**Figure 3 materials-14-05663-f003:**
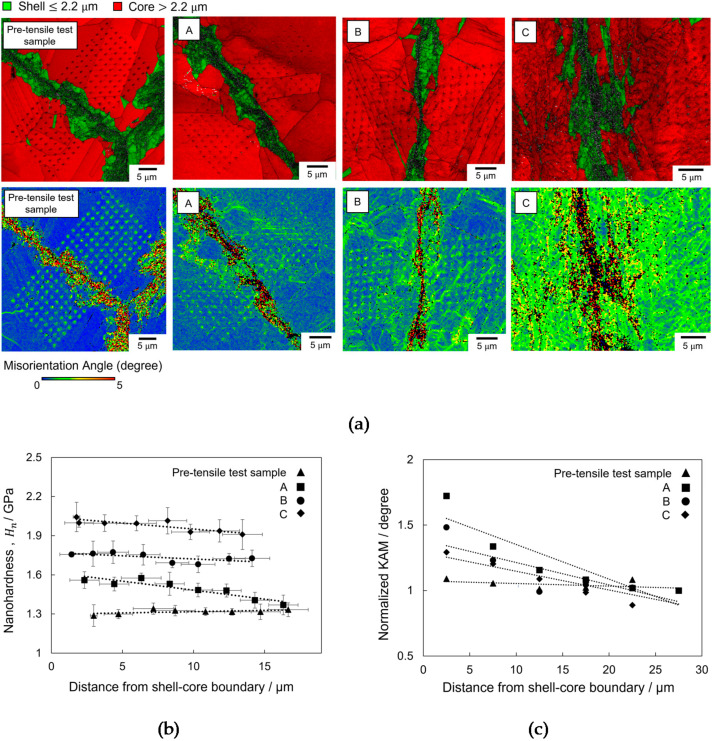
(**a**) IQ maps overlaid on the grain size (top) and KAM (bottom) maps with a 0.2 µm step size at the nanoindentation area of the pre- and post-tensile test samples. The arrayed dots represent the nanoindentation marks. (**b**) Nanohardness of the pre-tensile test sample and three positions labeled as A, B and C on the post-tensile test sample plotted as a function of the distance from the shell–core boundary. (**c**) Normalized KAM as a function of distance from the shell–core boundary for the pre-tensile test and the three positions labeled as A, B and C on the post-tensile test sample.

**Figure 4 materials-14-05663-f004:**
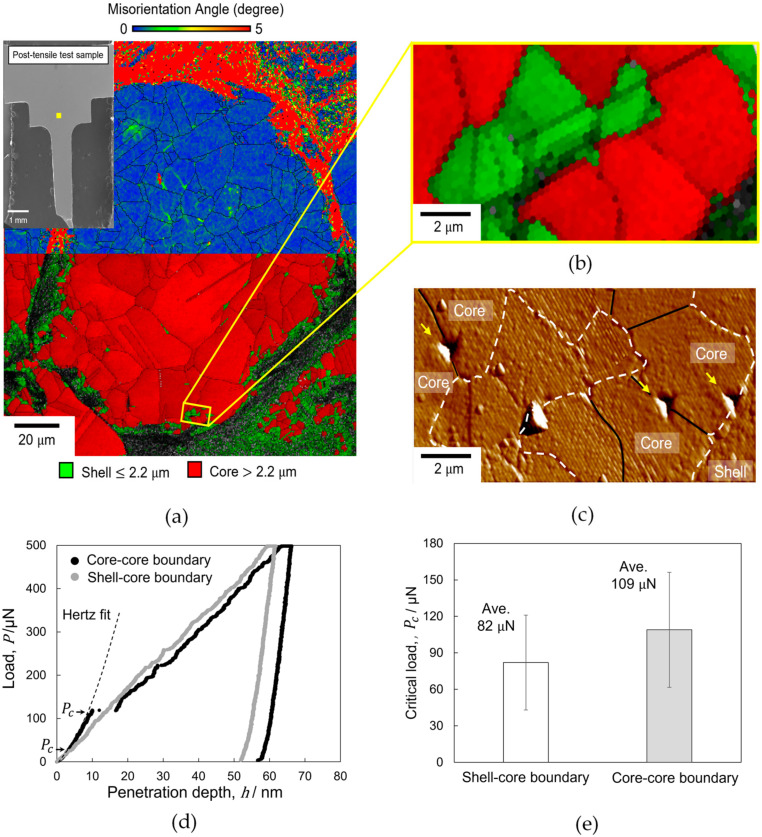
(**a**) KAM map (top) and IQ map overlaid with the color-coded map indicating the shell and core regions (bottom) with a 0.5 µm step size. The inset shows the observed position on the post-tensile test sample. (**b**) Magnified image of the shell and core region in the yellow box in (**a**) with higher resolution of EBSD scan step size of 0.2 µm. (**c**) Scanning probe microscopy image of the area shown in (**b**) after nanoindentation at the core–core and shell–core boundaries, which are indicated by yellow arrows. The white dotted line indicates the shell–core boundary. (**d**) Nanoindentation load–displacement curves of the core–core and shell–core boundaries with arrows indicating the critical load, $P_{c}$, for pop-in phenomena. (**e**) Bar graph of the average $P_{c}$ of the core–core and shell–core grain boundaries.

**Figure 5 materials-14-05663-f005:**
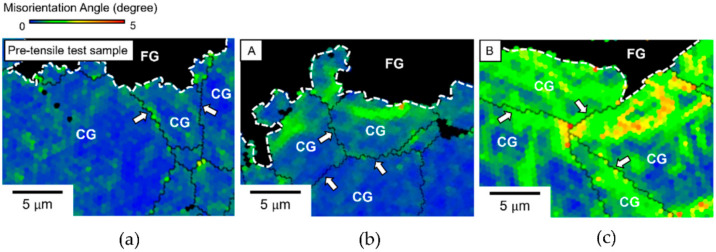
Magnified KAM maps at the shell–core boundary with a 0.5 µm step size. FG and CG refer to the fine grains in the shell region and the coarse grains in the core region, respectively. The black regions in each figure referred to the FG which are not indexed. (**a**) Before the tensile test, the KAM is nearly homogenous. (**b**) After the tensile test, a higher KAM can be observed near the shell–core boundary than at the core–core boundary at position A. (**c**) At position B, a higher KAM can be observed at both the shell–core and core–core boundaries. The shell–core boundary is highlighted by the dashed white line, and the core–core boundaries are indicated by white arrows.

**Figure 6 materials-14-05663-f006:**
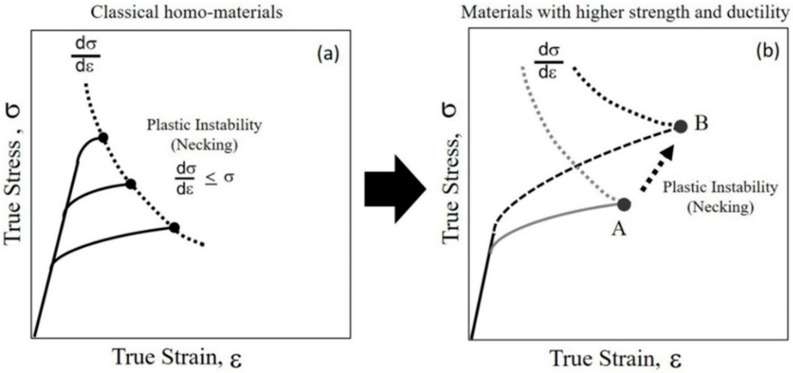
Schematics of true stress–true strain curves for (**a**) conventional homogenous materials and (**b**) high strength and high ductility materials [9].

## Data Availability

Data is contained within the article.

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
