# Peer review of "Local Deformation Behavior of the Copper Harmonic Structure near Grain Boundaries Investigated through Nanoindentation"

_materials, 2021, doi:10.3390/ma14195663_

Round 1
Reviewer 1 Report
The subject of the paper is interesting and well presented. Contribution to the investigation of the mechanical behavior of the bimodal harmonic structure is of great importance.
Some minor improvements and clarification in the methods part are needed. It is written in line 95 that KAM was obtained both with the step 0.4 and 0.02 μm. It is true for the fig1 that EBSD maps were obtained for both step sizes but what with others presented SEM/EBSD KAM maps, which step size was chosen, how the step size influence the KAM maps? ( obviously, it influences) and what KAM parameters were chosen in the TSL software? In fig. 5, it looks like the FG were not indexed or have a low confidence index? It needs to be explained in the materials and methods section how all the SEM/EBSD maps were analyzed. However, the overall assessment of the paper is high and I recommend the paper to publish.
Reviewer 2 Report
The paper present a complete communication about Local Deformation Behavior of the Copper Harmonic Structure
Near Grain Boundaries Investigated through Nanoindentation. Some changes need to be perfomed in order to consider to publication.
Authors should change the term nanohardness which is not correct. In fact they use the correct one throughout the text and they use this one too.
What do the authors mean by KAM value? There is no such value. You have to specify better what they mean.
